# Recent Advancement of Medical Patch for Transdermal Drug Delivery

**DOI:** 10.3390/medicina59040778

**Published:** 2023-04-17

**Authors:** Won Fen Wong, Kuan Ping Ang, Gautam Sethi, Chung Yeng Looi

**Affiliations:** 1Department of Medical Microbiology, Faculty of Medicine, Universiti Malaya, Kuala Lumpur 50603, Malaysia; wonfen@um.edu.my; 2Department of Medical Microbiology, University Malaya Medical Center, Kuala Lumpur 59100, Malaysia; angkp@ummc.edu.my; 3Department of Pharmacology, Yong Loo Lin School of Medicine, National University of Singapore, Singapore 117600, Singapore; 4School of Biosciences, Faculty of Health & Medical Sciences, Taylor’s University, Subang Jaya 47500, Malaysia

**Keywords:** transdermal, drug delivery, medical patch, development and technology

## Abstract

Transdermal patches are a non-invasive method of drug administration. It is an adhesive patch designed to deliver a specific dose of medication through the skin and into the bloodstream throughout the body. Transdermal drug delivery has several advantages over other routes of administration, for instance, it is less invasive, patient-friendly, and has the ability to bypass first-pass metabolism and the destructive acidic environment of the stomach that occurs upon the oral ingestion of drugs. For decades, transdermal patches have attracted attention and were used to deliver drugs such as nicotine, fentanyl, nitroglycerin, and clonidine to treat various diseases or conditions. Recently, this method is also being explored as a means of delivering biologics in various applications. Here, we review the existing literatures on the design and usage of medical patches in transdermal drug delivery, with a focus on the recent advances in innovation and technology that led to the emergence of smart, dissolvable/biodegradable, and high-loading/release, as well as 3D-printed patches.

## 1. Introduction

Transdermal drug delivery is an alternative way of delivering drugs via the skin layer [1,2]. The drug is carried through the skin into the bloodstream and circulates systemically in the body before reaching the target site [1,2]. The transdermal drug delivery method has several advantages over other routes of administration. Examples include the ability to deliver continuous doses of drugs over an extended period of time, the ability to bypass the digestive system, and the ability to avoid first-pass metabolism in the liver [3]. Other drug administration routes, such as intravenous, can cause pain and increase the risk of infection. Nevertheless, the oral route is inefficient, and in the inhalation method, it is difficult in controlling the dosage. In view of its advantages over other routes, transdermal administration is commonly used to deliver drugs for conditions such as smoking cessation, chronic pain, and motion sickness, as well as hormone replacement therapy [4,5,6].

A transdermal patch is a medicated patch that can deliver drugs directly into the bloodstream through the layers of the skin at a prescribed rate. In fact, patches are the most convenient method of administration. They are non-invasive, and treatment can last for several days and can be stopped at any time (Table 1). They come in different sizes and contain multiple ingredients. When applied to the skin, the patch can deliver active ingredients into the systemic circulation via diffusion processes. Transdermal patches may contain high doses of active substances that remain on the skin for an extended period of time. One of the first transdermal patches developed in 1985 was the nitroglycerin patch. The patch, developed by Gale and Berggren, uses a rate-controlling ethylene vinyl acetate membrane. Currently, several drugs are available as transdermal patches, including estradiol, clonidine, fentanyl, nicotine, scopolamine (hyoscine), and estradiol with norethisterone acetate (Table 2). The site of application may vary depending on the therapeutic category of the drug [7]. For example, nitroglycerin can be applied around the chest and estradiol around the buttocks or abdomen. The duration of drug release also varies depending on the usage, from the shortest (up to 9 h) to the longest (up to 9 days).

## 2. Transdermal Patch Design

Transportation of drug across the skin is affected by various factors, such as skin permeability, area, and duration of application, as well as metabolic activity of the skin (i.e., first pass metabolism). In fact, every drug has its unique properties, which can affect transdermal delivery. To achieve adequate skin absorption and penetration, the drug should be non-ionic and relatively lipophilic to cross the skin barrier. Molecules larger than 500 Daltons make it difficult to cross the stratum corneum, and ideally the therapeutic dose of the drug should also be less than 10 mg per day. 

## 3. Basic Component of Transdermal Patch

Transdermal patches typically consist of several layers that are designed to deliver the medication through the skin and into the bloodstream. Figure 1 illustrates the basic component of a medicated patch. The specific composition and structure of the patch may vary depending on the drug being delivered and the desired rate of drug release.

The backing layer is the outermost layer of the patch and serves to protect the other layers from the environment. This layer is usually made of a flexible, waterproof material such as polyethylene or polypropylene. The adhesive layer serves to attach the patch to the skin and keep it in place. It usually consists of a strong, hypoallergenic adhesive that is gentle on the skin. The drug layer contains drugs that are delivered through the skin. It is formulated to release the drugs at a constant rate over a period of time. The rate-controlling membrane serves to control the rate at which the drugs are released from the patch. Membranes are usually made of semi-permeable materials that allow the drugs to pass through the membrane at a controlled rate. Linen acts as a protector for the patch and adhesive. The patch must be removed before being applied to the skin surface.

## 4. Types of Transdermal Patches

In general, there are four main type of transdermal medical patches (drug-in adhesive, reservoir, matrix, and micro-reservoir systems), as shown in Figure 2. Most commercially available patches are categorized as reservoir or matrix systems [56]. 

### 4.1. Drug-in-Adhesive System

This is the simplest form of membrane permeation control system. The adhesive layer in this system contains drugs and serves to glue the different layers together. The drug mixture is sandwiched between the liner and backing.

### 4.2. Reservoir System

In this system, the drug reservoir is held between the backing layer and the rate-controlling membrane, and the drug is released through the microporous rate-controlling membrane. The drug can be in solution, suspension, or gel form, or can be dispersed in a solid polymer matrix within the reservoir chamber.

### 4.3. Matrix System

Drugs are uniformly dispersed in hydrophilic or lipophilic polymer matrices. The resulting drug-containing polymer is affixed to drug-containing discs of controlled thickness and surface area.

### 4.4. Micro-Reservoir System

This system is a combination of reservoir and matrix dispersion system. Here, the drug is prepared by first suspending drug solids in an aqueous solution of a water-soluble liquid polymer and then uniformly dispersing the solution in a lipophilic polymer to create thousands of non-leaching microscopic drug reservoirs. 

## 5. Microneedle-Based Patches

There are several types of microneedles, each with unique features and characteristics, as shown in Table 3. Overall, four major types of microneedle-based patches have been developed, namely solid, hollow, dissolving, and coated microneedle (Figure 3). The choice of microneedle type depends on the specific application and requirements of the user. 

Solid Microneedles: These are the simplest type of microneedles, consisting of solid needles that penetrate the skin to create tiny channels. Solid microneedles are commonly used for drug delivery and cosmetic treatments.Hollow Microneedles: These microneedles have a hollow core that allows for the delivery of fluids or drugs into the skin. Hollow microneedles are often used for transdermal drug delivery and sampling of interstitial fluid.Coated Microneedles: These microneedles have a coating that dissolves upon penetration of the skin, allowing for the release of drugs or other substances. Coated microneedles are often used for transdermal drug delivery.Dissolving Microneedles: These microneedles are made of materials that dissolve in the skin, allowing for the controlled release of drugs or other substances. Dissolving microneedles are often used for vaccines and other drug delivery applications.

## 6. Recent Advancement of Transdermal Patch

Traditional transdermal patches serve only two purposes: storage and release of drugs. While this method has some advantages, traditional patching has many challenges and drawbacks, for example limited dosage or low release. To date, there have been several advances in the field of transdermal drug delivery. These include the design of novel patches, which include the ability to sense and release drugs accurately, higher loading, and enhanced penetration and release of drugs. Overall, the field of transdermal drug delivery is an active area of research and development, with many exciting new developments on the horizon, as discussed below.

### 6.1. Smart Patches 

Smart patches are equipped with sensors and other technologies that can monitor patient conditions and adjust drug delivery accordingly. In 2014, a group of researchers developed a microneedle-based smart patch sensor platform for painless and continuous intradermal glucose measurement for diabetics. This patch uses a conducting polymer such as poly (3,4-ethylenedioxythiophene) (PEDOT) as an electrical mediator for glucose detection and as an immobilizing agent for the glucose-specific c-enzyme glucose oxidase (GOx) [72]. Further research and development resulted in a smart insulin releasing patch consisting of 121 microneedles containing nanoparticles. The patch painlessly penetrates into the interstitial fluid between subcutaneous skin cells. The nanoparticles in each needle contain insulin and the glucose-sensing enzyme glucose oxidase, which converts glucose into gluconate. These molecules are surrounded by hypoxia-responsive polymers. As shown in Figure 4, increased glucose oxidase activity in response to increased glucose creates an oxygen-depleted environment within the nanoparticles, which is sensed by the hypoxia-responsive polymer, triggering nanoparticle degradation and insulin release [73,74].

Wound healing is a complex and dynamic regenerative process with constantly changing physical and chemical parameters. Its management and monitoring offer great benefits, especially for bedridden patients. Iversen et al. reported an inexpensive, flexible, fully printed smart patch on the skin to measure changes in wound pH and fluid volume. Such bendable sensors can also be easily incorporated into wound dressings. The sensor consists of various electrodes printed on a polydimethylsiloxane (PDMS) substrate for pH and humidity measurements. The generated sensor patch has a sensitivity of 7.1 ohm/pH to the wound pH value. Hydration sensor results showed that the water content of the semi-porous surface can be quantified by the change in resistivity [75].

Besides wound healing, scientists have also developed a smart patch to monitor and treat diabetic foot ulcers (DFU). This system is fabricated from conductive hydrogel patches with a ultra-high transparency polymer network. Importantly, highly transparent conductive hydrogel patches can be used to visually monitor wound healing status, promote haemostasis, improve cell-to-cell communication, prevent wound infection, promote collagen deposition, and improve vascularity. By promoting angiogenesis, it effectively promotes the healing of DFU. In addition, the versatile intelligent patch can also achieve indirect blood glucose monitoring by detecting glucose levels in wounds, and timely detect movements of various sizes of human bodies. Interestingly, this smart patch can monitor chronic wound dressings and treat wounds at the same time [76].

Smart patches are also used to deliver natural compounds such as curcumin. The material consists of paraffin wax and polypropylene glycol as a phase change material (PCM). PCM was combined with graphene-based heating elements obtained by the laser scribing of polyimide films. This arrangement offers a new approach to smart patches whose release can be electronically controlled, and which allows repeated dosing. Emission is induced and terminated by controlled heating of the PCM rather than relying on passive diffusion, and permeation only occurs when the PCM transitions from solid to liquid. Curcumin delivery yields were found to be good and satisfactory [77].

### 6.2. Dissolving/Degradable Patches 

These patches are designed to dissolve on the skin and do not need to be removed and discarded. In general, these patches are made from biodegradable materials that are absorbed by the body after use. In a proof-of-concept paper published in 2019, researchers successfully administered the antibiotic gentamicin via a dissolving patch in a mouse model of bacterial infection [78] (Figure 5). The results showed that a gentamicin-dissolving microarray patch applied to mouse ears could control *Klebsiella pneumoniae* infection. In addition, mice treated with lysing patches had reduced bacterial burden in nose-associated lymphoid tissue and lungs compared with untreated control.

Dissolving microneedles (MNs) show high efficiency in the delivery of poorly permeable drugs and vaccines. A two-step injection and centrifugation process was used to localize insulin to the needle and achieve efficient transdermal delivery of insulin. The relative pharmacological availability and relative bioavailability (RBA) of insulin from MN patches were 95.6% and 85.7%, respectively. This study demonstrates that the use of dissolving patches for insulin delivery achieves a satisfactory relative bioavailability (RBA) compared to conventional subcutaneous injection, demonstrating the effectiveness of dissolving patches for diabetes treatment [79].

On the other hand, scientists have developed a biodegradable microneedle patch that delivers hyaluronic acid (HA) antigen-peptide conjugates for prophylactic cancer immunotherapy [80]. A cytotoxic T-cell epitope peptide (SI-INFEKL) is conjugated to HA-loaded biodegradable HA microneedle (MN) patches to efficiently deliver antigens to the skin’s immune system. Interestingly, a single transdermal vaccination with an MN patch containing the HA-SIINFEKL conjugate significantly increased tumor growth in B16 melanoma model mice by enhancing antigen-specific cytotoxic T-cells.

Another research group developed a hypotensive biodegradable patch for transdermal delivery of sodium nitroprusside (SNP) in combination with sodium thiosulfate (ST) [81]. Dissolvable microneedles loaded with SNPs and STs were fabricated by centrifugal casting. In this method, SNPs were stably packaged into microneedles and immediately released into the systemic circulation. Antihypertensive microneedle therapy (aH-MN) achieved rapid and strong blood pressure reduction. It met the clinical requirements for blood pressure management in hypertensive emergencies. Concomitant administration of ST effectively suppressed side effects (e.g., organ damage) caused by the continuous intake of SNPs. This study presented an efficient and patient-friendly biodegradable patch for antihypertensive therapy.

Transdermal patches are also commonly used in the cosmetic industry. However, the non-degradable polymers used in cosmetic patches are of concern because they can pollute the environment when disposed irresponsibly in open areas. In one study, biodegradable polylactic acid (PLA) was recommended due to its lack of toxicity. The results showed that the PLA/phycocyanin-alginate composite made with a phycocyanin/alginate ratio of 40/60 at 20 °C for 20 h had the best properties in terms of film flexibility and release properties [82]. Overall, the results are promising but warrant further in vivo or clinical study for further development.

### 6.3. Three-Dimensional (3D)-Printed Patches 

Researchers are using 3D printing technology to create customized transdermal patches that can be tailored to the individual needs of each patient [83]. One good example is the use of a 3D-printed patch for wound healing. In a study by Jang et al., gelatin methacrylate (GelMA) was tested as a viable option with tunable physical properties. GelMA hydrogel incorporating a vascular endothelial growth factor (VEGF)-mimicking peptide was successfully printed using a 3D bio-printer owing to the shear-thinning properties of hydrogel inks. The 3D structure of the hydrogel patch had high porosity and water absorption properties. VEGF peptide, which is slowly released from hydrogel patches, can promote cell viability, proliferation, and tubular structure formation, indicating that the 3D Gel-MA-VEGF hydrogel patch can be used for wound dressing [84].

On the other hand, a three-dimensional (3D) printing technique called continuous liquid interface production (CLIP) was used to design and fabricate transdermal patches. The multifaceted microneedle design increased the surface area compared to the smooth square pyramid design, ultimately resulting in the improved surface coating of model vaccine components (ovalbumin and CpG). In the study, they used fluorescent tags and live animal imaging to assess in vivo charge retention and bioavailability in mice as a function of delivery route. Compared with subcutaneous bolus injection of soluble components, transdermal administration not only resulted in improved skin charge retention, but also improved the activation of immune cells in draining lymph nodes. Moreover, the delivered vaccine elicited a strong humoral immune response with higher total IgG (immunoglobulin G) and a more balanced IgG1/IgG2a repertoire, resulting in dose sparing. Furthermore, it elicited a T-cell response characterized by functional cytotoxic CD8+ T-cells and CD4+ T-cells secreting Th1 (T helper type 1) cytokines. In short, CLIP 3D-printed microneedles coated with vaccine components provide a useful platform for non-invasive, self-administered vaccination [85].

Another group of researchers designed and printed the patch using stereolithography (SLA) technology with a proprietary class I resin. They showed that these patches can be used for transdermal delivery of high molecular weight antibiotics such as rifampicin (M(w) 822.94 g/mol). This drug suffers from gastric chemical instability, reduced bioavailability, and severe hepatotoxicity. The patch was engineered with sub-apical holes present in one-quarter of the needle tip to enhance the mechanical strength and integrity of the patch array. The tips were characterized by optical and electron microscopy to determine print quality and uniformity across the array. The system also underwent mechanical characterization for failure and penetration analysis. The authors systematically evaluated the ex vivo penetration and consequent transport of rifampicin through porcine skin. Moreover, in vivo study of rifampicin administration via the 3D-printed patch demonstrated efficient penetration and desirable bioavailability [86].

Another technique known as powder extrusion (DPE) has emerged as the most viable approach due to its ability to directly process excipients and pharmaceuticals in one step [87]. The study was set to determine whether different grades of ethylene-vinyl acetate (EVA) copolymers could be used as new starting materials for the fabrication of transdermal patches. By choosing two model drugs with different thermal behavior (i.e., ibuprofen and diclofenac sodium), they also wanted to consider the versatility of this EVA excipient in manufacturing patches for custom transdermal therapy. EVA was combined with 30% (w/w) of each model drug. Fourier transform infrared (FT-IR) spectra confirmed that the starting material was effectively incorporated into the final formulation, and thermal analysis revealed that the extrusion process changed the crystalline morphology of the raw polymer, leading to increased crystallization at smaller thicknesses. This study indicated that EVA and direct powder extrusion technology may be promising tools for the fabrication of transdermal patches. By choosing an EVA type with appropriate VA content, it is possible to print drugs with different melting points while maintaining thermal stability. Furthermore, the desired drug release and permeation profiles of drugs can be achieved. In fact, this is an important advantage from the point of view of personalized medicine.

Lim et al. reported the use of 3D printed personalized patches that conform to the skin surface for Acetyl-hexapeptide 3 (AHP-3) delivery. However, commercially available photocurable resins for 3D printing are not suitable for fabricating drug-loaded delivery systems. In this study, two liquid monomers, namely polyethylene glycol diacrylate (PEGDA) and vinylpyrrolidone (VP) in different proportions were used to improve the mechanical strength, polymerization rate, and swelling rate of the final polymer. Optimal drug loading on the resin indicated that AHP-3 remained stable throughout the manufacturing process and had no effect on the physical properties of the final polymer. Using a 3D scanned facial model, a personalized patch was designed in CAD (computer-aided design) software and fabricated in optimized resin using a digital light processing (DLP) 3D printer. In vitro characterization of the prepared transdermal patches showed their ability to penetrate human cadaver skin, and they remained intact after compression. The final polymer was also minimally cytotoxic to human dermal fibroblasts. This is the first study demonstrating personalized patches made using photopolymers and may be a novel approach to improve the transdermal delivery of drugs for effective wrinkle management [88].

### 6.4. High Loading/Release Patches 

Long-acting transdermal drug delivery requires high drug loading and controlled drug release. In order to improve drug-polymer miscibility and achieve controlled drug release, a novel hydroxyphenyl (HP)-modified pressure-sensitive adhesive (PSA) was developed [89]. The results show that the dual-ionic H-bonds between R(3)N and R(2)NH-type drugs and HP-PSA are reversible and relatively strong, unlike ionic and neutral H-bonds. This allowed patches to significantly increase the drug loading from 1.5- to 7-fold and control the drug release rate from 1/5 to 1/2 without changing the overall release profile. Pharmacokinetic results showed that the HP-PSA-based high-load patch achieved sustained drug concentrations in plasma, avoided sudden release, increased area under the concentration-time curve (AUC), and average dwell time by more than 6x, indicating the potential for long-acting drug delivery. In addition, its safety and mechanical properties are met. Mechanistic studies have shown that repulsion of ionic drugs in HP-PSA increases drug loading, and relatively strong interactions can also control drug release. Incomplete hydrogen bond transfer determined its reversibility, making the percentage drug release equal to that of non-functional PSA. In short, the high drug loading efficiency and controlled drug release capability of HP-PSA and its unique interactions will contribute to the development of long-acting transdermal drug delivery systems. Moreover, the construction of double-ionic H-bonds provides further inspiration for various drug delivery systems in non-polar environments.

Pharmaceutical polymers are widely used to inhibit drug recrystallization through strong intermolecular hydrogen and ionic bonding, but at the expense of drug release rates in transdermal patches. To overcome this difficulty, a group of researchers came up with the idea of using a new ionic liquid (drug IL) strategy to increase drug loading [90]. A carboxyl-based pressure sensitive adhesive (PSA) was chosen as a model polymer. The results showed a five-fold increase in PSA drug load. This was caused by the synergistic effect of strong ionic and normal hydrogen bonds formed between the carbonyl groups of the drug and PSA. This study demonstrated an entirely new mechanism of action and provided a powerful tool for the development of high-drug load, high-release patches. In another study, the same group of researchers constructed a high-capacity, high-release transdermal patch with COOH polyacrylate polymer (PA-1) to deliver non-steroidal anti-inflammatory drugs (NSAIDs), namely ibuprofen. The drug load and skin absorption of PA-1 were improved by 2.4-fold and 2.5-fold, respectively. The hydrogen bond formed between the drug (COOH) and PA-1 (COOH) is weakened by repulsive interactions, whereas the enhanced conductivity of PA-1 was confirmed by dielectric spectroscopy, electron paramagnetic resonance (EPR) spectra, four-point probe method, and molecular modeling with the appearance of COO-. In summary, these results showed that ion–ion repulsion by reducing hydrogen bonding can be a viable way to build large-capacity, high-emission patches [91].

## 7. Potential Application of Transdermal Patches

Previously, we listed a number of therapeutically active drugs marketed as transdermal patches in Table 2. As technology and research have advanced, numerous potential application areas for transdermal patches have been explored, as described below.

### 7.1. Transdermal Patches for Patches for Vaccination 

Researchers are developing transdermal patches that can deliver vaccines through the skin, potentially offering a more convenient and less painful alternative to injections. A good example is the microneedle-based smallpox vaccine patch. When this vaccine patch was applied to mice, neutralizing antibodies were induced from 3 weeks after immunization. Levels were maintained for 12 weeks, and there was a significant increase in IFN-γ secreting cells, suggesting that the transdermal patch could serve as an alternative delivery system for vaccination and preservation [92].

Another research group designed a lytic microneedle patch for influenza vaccination that targets skin antigen-presenting cells. Microneedles were created using a biocompatible polymer that encapsulates an inactivated influenza virus vaccine for insertion and dissolution into the skin within minutes. The patch elicited strong antibody and cell-mediated immune responses in mice that provided complete protection against lethal challenge. The results provide a new technique for simpler and safer vaccination with improved immunogenicity by using a transdermal patch, potentially enabling increased vaccination coverage [93].

### 7.2. Transdermal Patches for Gene Therapy 

Recently, transdermal patches have been investigated as a way of providing gene therapy to deliver genetic material to defective cells [94]. Pioneering studies sought to deliver genes and photothermic agents simultaneously to the cancer cells. For this purpose, transdermal patches co-loaded with p53 DNA and IR820 (a near infrared dye) were prepared by a two-step casting procedure. Hyaluronic acid was first constructed as the matrix, before p53 DNA and IR820 were primarily loaded onto the patches. The patches efficiently penetrated the stratum corneum and rapidly dissolved to release p53 DNA and IR820 at subcutaneous tumor sites. The patch showed an excellent anti-tumor effect in vivo due to the synergistic effect of gene therapy and photothermal agents. These findings demonstrate that a transdermal patch could be a promising strategy for subcutaneous tumor treatment [95].

### 7.3. Transdermal Patches for Insulin Delivery 

Transdermal insulin delivery patches are used to deliver insulin across the skin and into the bloodstream to treat diabetics. Insulin is a hormone secreted by the pancreas that plays a role in regulating blood sugar levels in the body. Diabetics can have high blood sugar levels because they cannot produce enough insulin or use the insulin produced by their body effectively. To date, several new techniques have been reported, including the use of ionic liquids, choline bicarbonate and geranic acid (CAGE), liposome, and nanomaterials to deliver insulin [96,97,98,99,100]. Transdermal patches for insulin delivery can provide a convenient and discreet alternative to traditional methods of insulin delivery, such as injections and insulin pumps [101]. The patches are typically applied to the skin on the abdomen, upper arm, or thigh and are designed to release a consistent dose of insulin over a specific period of time.

Of note, there are several challenges to overcome when developing transdermal patches for insulin delivery. Insulin is a large protein molecule that is not easily absorbed through the skin. To overcome this, researchers have developed a new approach for transdermal protein delivery using a water-swellable spherical double-layered microneedle (MN) patch at the tip. This design allows MNs to mechanically engage soft tissue through selective distal swelling after skin insertion. Furthermore, long-term release of loaded proteins was achieved by passive diffusion through the swollen tips. Insulin-loaded MN patches released 60% insulin upon immersion in saline for 12 h, and approximately 70% of the released insulin appeared to retain its structural integrity. Animal studies have shown that insulin release from swelling MN patches is prolonged, leading to a gradual decrease in blood glucose levels [102]. Overall, transdermal patches for insulin delivery have the potential to provide a convenient and effective method to deliver insulin to diabetic patients. However, further research and development is needed to optimize the efficacy and safety of these products.

### 7.4. Transdermal Patches for Cardiovascular Diseases 

In a heart failure scenario, pharmacokinetics (PK) and pharmacodynamics (PD) are frequently altered to accommodate hypoperfusion systemic conditions due to reduced cardiac ejection fraction [103]. In addition, drug metabolism and metabolite clearance are reduced in the renal failure [104]. Moreover, hypoalbuminemia and hepatic congestion due to heart failure impair drug absorption [105]. Therefore, transdermal patch delivery systems provide a drug delivery solution. For example, propranolol is a nonselective beta-adrenergic blocker. Its hepatic first-pass metabolism is highly altered when taken orally, with a bioavailability of approximately 23% [106,107]. A result of a previous animal study with rabbits showed that oral propranolol gave a Cmax of 56.4 ng/mL within 13.2 min. However, due to liver metabolism involvement, its bioavailability was 12.3% [107]. On the other hand, the transdermal propranolol patch achieved a steady-state plasma concentration (Css) of 9.3 ng/mL after 8 h of initial lag time, recording a bioavailability of 74.8% higher than oral propranolol [107].

On the other hand, Bisono^®^ Tape is a transdermal patch that formulates with bisoprolol as the active ingredient [108] to manage aortic dissection [109], premature ventricular contraction [110], orthostatic hypotension due to heart failure [111], and atrial fibrillation [112]. A comparative study of edematous and non-edematous patients with the Bisono^®^ Tape 4 mg patch showed that the Cmax of the oedema group was 13.3 ng/mL, and the Cmax of the non-oedema group was 17 ng/mL [113]. This study set out to clarify the effect of systemic edema on the absorption of beta-blockers from skin patches in critically ill patients. However, they discovered that blood levels and the heart rate-lowering effects of bisoprolol after application of the bisoprolol skin patch is not affected by systemic edema of the lower extremities.

Another antihypertensive drug that uses transdermal patch delivery is clonidine. Clonidine is an α 2-adrenergic agonist initially used to treat hypertension [114]. It was used for purposes such as attention-deficit hyperactive disorder (ADHD) [115] and drug withdrawal syndrome [116]. The transdermal clonidine patch was introduced in 1983 and was approved by the FDA in 1984 [117,118]. Since then, a comparative study of oral and transdermal clonidine has been conducted [119]. The results showed no difference in Cmax between oral clonidine (0.39 ng/mL) and transdermal clonidine (0.3 ng/mL), but the half-life of transdermal clonidine was longer than oral clonidine (31.9 h vs. 10.8 h) [119]. Furthermore, they showed no difference in the antihypertensive effect [119]. Losartan, an angiotensin II receptor blocker (ARB), is also being developed for transdermal drug delivery. Previously, a rat skin study with proniosome transdermal drug delivery was designed and studied. Transdermal losartan was shown to contribute to a Cmax of 141 ng/mL and 152 ng/mL when taken orally. However, the bioavailability of transdermal losartan is 1.93-fold that of oral losartan [120]. 

Nitroglycerin is another drug worth mentioning in cardiovascular therapy. Lauder Blanton used nitroglycerin to relieve angina pectoris and first noted drug resistance after repeated doses in 1867 [121]. Ferid Murad found that nitric oxide (NO) from nitroglycerin acts on vascular smooth muscle by activating cyclic guanosine monophosphate (cGMP), resulting in vasodilation [122]. The first transdermal nitroglycerin patch was developed by Gale and Berggren (Patent access US-4615699-A) in 1985. A year later, a two-way crossover study was performed on twenty-five healthy males with Nitro-Dur and another type of nitroglycerin transdermal patch, Nitro-Dur II, which showed an average Cmax of 0.383 ng/mL and 0.432 ng/mL, respectively [123].

### 7.5. Transdermal Patches for Hormonal Deficiencies and Contraception 

Attempts to deliver hormones transdermally can be traced back to 1938, when an attempt was made to apply a testosterone-containing ointment to castrated male guinea pigs. Since then, the cutaneous application of estrone and follicle-stimulating hormone to treat amenorrhea has been investigated. The first estradiol transdermal patch was introduced as a reservoir in 1984. A 0.05 mg/day patch applied twice weekly for 3 weeks showed a mean steady-state plasma estradiol concentration (Css) of 38 ng/L (0.038 ng/mL) [124]. Menorest^®^ was then developed using a transdermal matrix delivery system [125,126]. The matrix transdermal delivery system offers a better pharmacokinetic profile, lesser plasma estradiol fluctuation, and improved local tolerability upon application [127]. Menorest^®^ 50 had a Cmax of 51 pg/mL (0.051 ng/mL) upon its steady state [128]. Then came a new version of the estradiol transdermal matrix patch called Climara. At a nominal dose of 50 µg in 24 h, Climara showed a Cmax of 98 pg/mL (0.098 ng/mL), compared to Menorest’s Cmax (87 pg/mL or 0.087 ng/mL) upon the steady state [129]. However, Menorest has a higher absorption rate and a shorter time to reach the maximum concentration (Tmax) [129]. 

Ethinylestradiol is another estrogenic drug used for contraception [130]. In November 2001, the FDA approved Ortho Evra™, the first transdermal ethinyl estradiol contraceptive. It is a combination of norelgestromin and ethinyl estradiol [131]. A related pharmacokinetic study showed that transdermal ethinylestradiol reached a Cmax of 58.7 to 71.2 pg/mL, with a minimum half-life of 16.1 upon different sites of the application field [131]. Through early studies on the efficacy of ethinylestradiol transdermal patches, drug compliance was much better compared to oral pills statistically [132].

On the other hand, testosterone has been used to treat male hypogonadism [133]. There are several ways to deliver testosterone, including intravenous testosterone enanthate and transdermal testosterone patches (reservoir and matrix types). Intravenous administration achieved a Cmax of over 1200 ng/L (1.2 ng/mL) 24 h post-dose, with a long half-life of 7–9 days (Drugbank access: DB13944). For reservoir testosterone transdermal patches such as Androderm, Cmax was 765 ng/L (0.765 ng/mL) with a mean Tmax of 8 h at 16 weeks of treatment [134]. Nevertheless, new matrix-type testosterone transdermal patches showed higher testosterone concentrations for delivery after 15–19.5 h (mean Cmax ranged from 4.33 to 6.18 ng/mL) [135]. Once the patch is removed from the skin, the mean half-life of testosterone is 1.3 h [135].

### 7.6. Transdermal Patches for Central Nervus System (CNS) Disorder

There are advantages in developing transdermal drug delivery systems for central nervous system-related drugs. First, it provides sustained therapeutic dosing at plasma levels. Second, the transdermal drug delivery system exhibited a favorable pharmacological profile and bioavailability. Third, patients tolerate it well, thus reducing systemic side effects [136]. The following discussion covers the pharmacokinetics of methylphenidate, rotigotine, selegiline, asenapine, donepezil, and rivastigmine for transdermal and nondermal approaches. 

Methylphenidate, known as Ritalin, has been used to treat ADHD. It blocks presynaptic norepinephrine (NE) and dopamine reuptakes, creating stimulant effects in precortex neurons [137]. Regarding the pharmacokinetics of d-enantiomer methylphenidate, the Tmax for methylphenidate tablets was 2.36 h (Cmax of 18.12 ng/mL) for immediate-release tablets and 1.95 h (Cmax of 20.75 ng/mL) for chewed sustained-release tablets [138]. The elimination half-life of methylphenidate was approximately 5 h (5.33–5.69 h) [138]. Methylphenidate transdermal patches achieved a mean Cmax ranging from 20.0 to 46.5 ng/mL, with a mean Tmax ranging from 7.12 to 8.78 h, based on the size of the patches [139]. Rotigotine is currently incorporated into transdermal patches to treat Parkinson’s disease [140]. This drug demonstrates dose-proportional pharmacokinetics with stable plasma concentrations over 24 h [141]. Through a pilot study, a single dermal dose of rotigotine (4 mg/24 h) had a mean Cmax of 0.56 ng/mL at 19 h and a terminal half-life of 5.3 h [142]. 

Another medication used to treat Parkinson’s disease is selegiline (brand name Emsam). It is a selective and irreversible monoamine oxidase inhibitor that targets monoamine oxidase B. However, it can inhibit monoamine oxidase A when there is a large dose, causing an increase in cerebral serotonin and norepinephrine [143]. The pharmacokinetics of selegiline tablets in the Parkinson’s disease population have been reviewed previously [144]. The Cmax is 0.9–2.2 ng/mL for 2 × 5 mg tablets after 0.6–0.9 h post oral administration [144,145]. For the transdermal selegiline patch, the Cmax is 2.1 ng/mL for the 1 selegiline transdermal system (STS) patch after 16.5 to 17.3 h post-administration. The elimination half-life for the transdermal selegiline patch is 27.6 to 36.6 h [144,145]. For the pharmacokinetics of selegiline among the major depressive disorder population, the Cmax was 2.162 ng/mL after 18.4 h post application of the transdermal selegiline 6 mg/24 h patch [146]. The elimination half-life was 20.1 h [146]. In addition, asenapine is another drug designed for schizophrenia and the maniac phase of bipolar disorder [8]. For sublingual asenapine in a steady state, the 5 mg BID dosage gave a Cmax of 4.23 ng/mL after 1.75 h. The 10 mg BID dosage gave a Cmax of 6.56 ng/mL after 1.96 h. Both shared a half-life of 17.7 h [9]. On the other hand, the Cmax of the transdermal asenapine patch was 1.14 to 4.68 ng/mL after 12 to 16 h, based on the patch formulation [9]. The half-life for the transdermal asenapine patch was 33.9 h [9]. 

There are three medications to treat Alzheimer’s disease, namely donepezil, galantamine, and rivastigmine [147]. They are cholinesterase inhibitors that enhance cholinergic transmission in mild to moderate Alzheimer’s disease [148]. Regarding the pharmacokinetics of these drugs, oral donepezil reached a Cmax of 3.2–11.6 ng/mL after 3.2–4.7 h [149]. The elimination half-life of oral donepezil ranged from 53.8 to 82.8 h [149]. The Cmax of the donepezil transdermal patch ranged from 5.24 to 20.36 ng/mL at 74 to 76 h in a dose-dependent manner, with elimination half-lives ranging from 63.77 to 94.07 h [20]. Studies have shown that the rivastigmine patch has a 20% lower Cmax and a 14 times longer Tmax than the oral solution. In this study, rivastigmine 3 mg oral solution produced a Cmax of 7.63 ng/mL in one hour on average. The elimination half-life was also 1.45 h [45,46]. Alternatively, a 9.5 mg/24 h rivastigmine transdermal patch achieved a Cmax of 5.84 ng/mL after a 14.1 h average, with an elimination half-life of 3.02 h [45,46]. For Galantamine, a 10 mg dose of galantamine tablet produced a Cmax of 49.2 ng/mL at 0.88 h and an elimination half-life of 5.68 h [150,151]. Although there are no existing transdermal galantamine patches on the market, several attempts have been made to develop one, for instance a novel patch TAH-8801 from TAHO Pharmaceuticals is currently undergoing Phase III trials. Apart from that, a group of researchers are developing pressure-sensitive adhesive patches to transdermally deliver galantamine continuously [152].

### 7.7. Transdermal Patches for Infectious Diseases 

Advances in transdermal drug delivery approaches unlock the potential for new drug delivery methods. There are currently efforts to try other drugs for transdermal delivery, such as antibiotics and vaccines. As for transdermal antibiotics, the zwitterionic characteristic cephalexin was incorporated into solid lipid nanoparticles (SLNs) to develop a transdermal cephalexin patch. This showed a stable antibiotic effect with minimal antibiotic use [153]. Another approach is to use bacterial cellulose/polycaprolactone (BC/PCL) patches to load amoxicillin, ampicillin, and kanamycin for transdermal delivery development. Such methods can produce a solid bactericidal effect against *Staphylococcus aureus* and *E. coli* [154]. Additionally, tetracyclines have been incorporated into hydrogel-forming microarray patches for transdermal delivery systems. An in vivo study in rats determined the Cmax of such an approach and showed 7.40 μg/mL at 24 h compared to the Cmax of oral tetracycline of 5.86 μg/mL at 1 h [155]. A similar approach was applies to vancomycin, in which the Cmax of the rodent model with a hydrogel-forming microarray patch was 3.29 μg/mL at 48 h post-treatment and the Cmax of the rodent model with a dissolving microarray patch showed 1.58 μg/mL at 24 h post-treatment, in comparison to oral with Cmax 3.37 μg/mL and intravenous with Cmax 50.34 μg/mL [156].

## 8. Conclusions and Future Challenges

Transdermal patch technology is a valuable drug delivery method with many advantages over other delivery routes. Patches can bypass the digestive system and first-pass metabolism to provide continuous dosing of drugs over an extended period of time. They are commonly used to deliver drugs for various indications such as chronic pain, motion sickness, and hormone replacement therapy. In recent years, there have been many advances in transdermal patch technology, including the development of smart, dissolving/biodegradable, high-loading/release and 3D-printed patches. Transdermal patches have the potential to provide a convenient and effective means of drug delivery for a variety of ailments, but some challenges lie ahead, such as the possibility of self-inflicted toxicity due to improper dosing, poor adhesion, low drug penetration, potential trigger for skin irritation, or patch failure. All of this warrants further research and development to optimize the safety and efficacy of this delivery system.

## Figures and Tables

**Figure 1 medicina-59-00778-f001:**
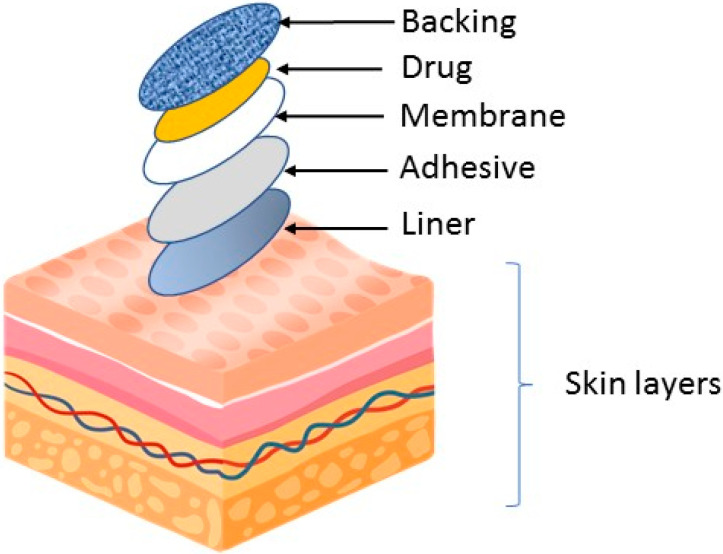
Basic component of a transdermal medical patch.

**Figure 2 medicina-59-00778-f002:**
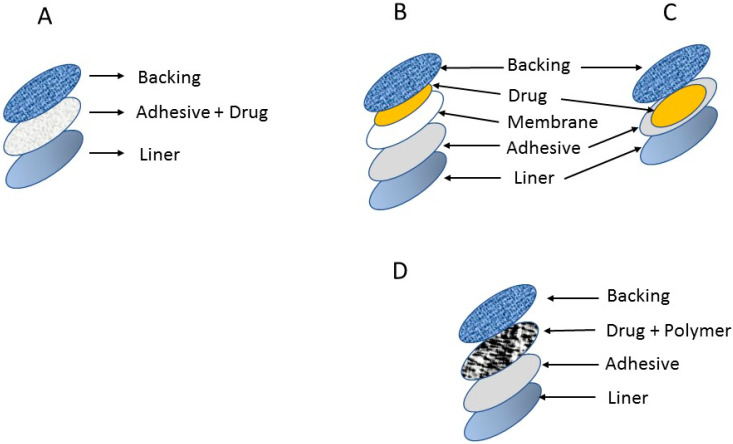
Types of transdermal patches: (**A**) drug-in adhesive system; (**B**) Reservoir system; (**C**) matrix system; (**D**) Micro-reservoir system.

**Figure 3 medicina-59-00778-f003:**
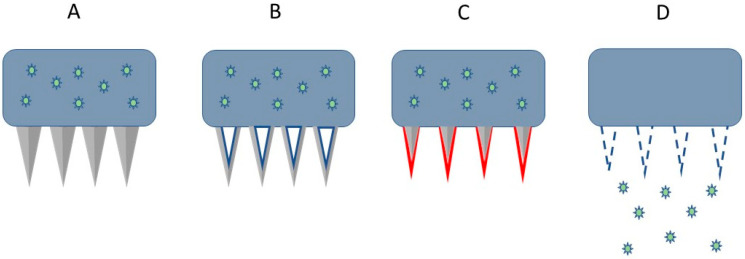
The microneedle-based patch: (**A**) solid; (**B**) hollow; (**C**) coated; (**D**) dissolving.

**Figure 4 medicina-59-00778-f004:**
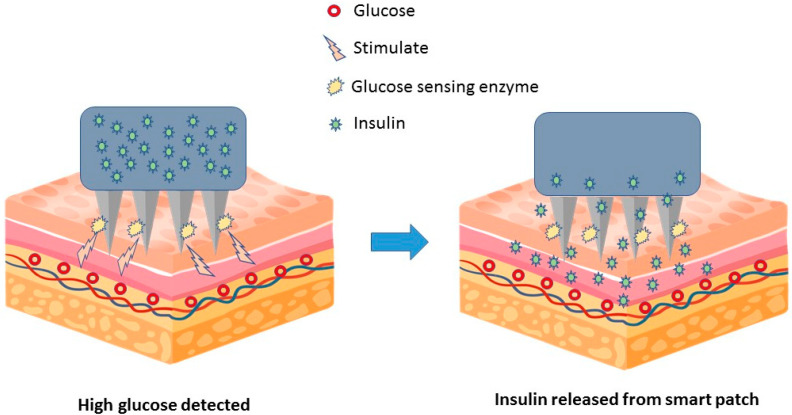
The microneedle-based patch painlessly penetrates into the skin layer. The smart patch contains insulin and the glucose-sensing enzyme glucose oxidase, which converts glucose into gluconate. Higher glucose oxidase activity in response to increased glucose triggers nanoparticle degradation and insulin release.

**Figure 5 medicina-59-00778-f005:**
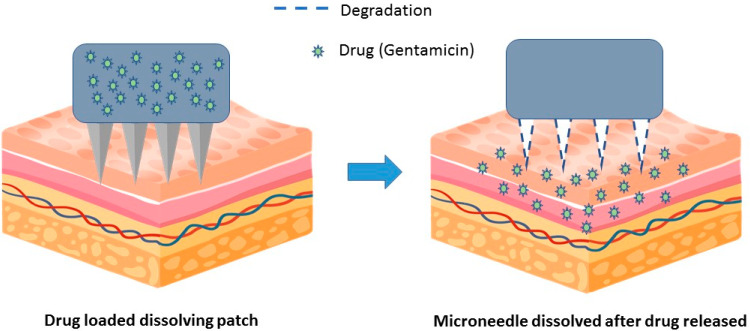
The microneedles in these patches are made out of biodegradable materials. After gentamicin has been released from the patch, the microneedle dissolve on the skin.

**Table 1 medicina-59-00778-t001:** Pros and cons of using transdermal patches.

Advantages	Disadvantages
Continuous dosing, multi-day treatment	Limited type of medication
Bypass the digestive system	Skin irritation
Avoid first-pass metabolism	Inconsistent absorption
Can be terminated anytime	Patch failure
Less invasive	Limited dosing option

**Table 2 medicina-59-00778-t002:** A summary of transdermal patches/products and their unique features.

Drugs	Indication	Product Name	Duration of Application	Reference
Asenapine	Mania, bipolar disorder	Secuado^®^	24 h	[8,9]
Bisoprolol	Atrial fibrillation	Bisono^®^	24 h	[10]
Buprenorphine	Management of pain	Butrans^®^	7 days	[11,12,13]
Clonidine	Hypertension, Tic disorder, Tourette syndrome, Attention deficit hyperactivity disorder (ADHD)	Catapres-TTS^®^	7 days	[14,15,16,17,18]
Dextroamphetamine	ADHD	Xelstrym^®^	Up to 9 h	[19]
Donepezil	Alzheimer disease	Adlarity^®^	7 days	[20,21]
Estrogen	Postmenstrual syndrome	Fematrix^®^	7 days	[22,23]
Ethinyl Estradiol	Prevent pregnancy	Ortho Evra^®^	7 days	[24,25]
Fentanyl	Moderate/severe pain	Duragesic^®^	72 hours	[26]
Granisetron	Anti-emetic	Sancuso^®^	Up to 7 days	[27,28,29]
Levonorgestrel, Estradiol	Postmenstrual syndrome	Climara Pro^®^	7 days	[30,31]
Lidocaine	Treatment of pain	Lidoderm^®^ Dermalid^®^	up to 3 times daily for no more than 12 hours	[32,33]
Methylphenidate	ADHD	Daytrana^®^	Up to 9 days	[34]
Nicotine	Smoking cessation	Habitrol^®^, Nicoderm^®^ Nicoderm CQ^®^ Nicorette^®^	24 h 16 h	[35,36,37]
Nitroglycerin	Angina pectoris Relieve pain after surgery	Minitran^®^ Nitro-dur^®^	12–14 h	[38,39,40,41]
Norethindrone Estradiol	Symptoms of menopause	Combipatch^®^	3–4 days	[42]
Oxybutynin	Overactive bladder	Oxytrol^®^	3–4 days	[43,44]
Rivastigmine	Alzheimer disease	Exelon^®^	24 h	[45,46]
Rotigotine	Parkinson’s disease	Neupro^®^	24 h	[47]
Selegiline	Depression	Emsam^®^	24 h	[48]
Scopolamine	Motion sickness	Transderm-scop^®^	72 h	[49,50]
Testosterone	Hypogonadism in males	Androderm^®^	24 h	[51,52]
17-β-Estradiol	Postmenstrual syndrome and osteoporosis	Alora^®^ Climara^®^ Estraderm^®^ Vivelle-Dot ^®^ Vivella^®^ Menostar^®^ Minivelle^®^	3–4 days 7 days 3–4 days 3–4 days 3–4 days 7 days 3–4 days	[53,54,55]

**Table 3 medicina-59-00778-t003:** Microneedle types with their unique features.

Type	Material	Structure	Use	Dose	Delivery Rate	References
Solid	Silicon, Metal, Polymer	Simple	Can be reuse	Small dose	Fast	[57,58,59]
Hollow	Silicon	Simple	Can be reuse	Large dose	Fast	[60,61,62,63]
Coated	Polymer, Sugar, Lipids	Complex	Single	More precise dosing	Fast	[64,65,66,67]
Dissolving	Polymer	Complex	Single	More precise dosing	Slow	[68,69,70,71]

## Data Availability

Not applicable.

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
