# Peer review of "Recent Advancement of Medical Patch for Transdermal Drug Delivery"

_medicina, 2023, doi:10.3390/medicina59040778_

Round 1
Reviewer 1 Report
This review provides an overview of conventional transdermal products as well as recent technologies for transdermal drug delivery. However, there are some areas that need to be revised before publication.
1. Table 2 does not cover recently approved transdermal patches such as asenapine, donepezil, rivastigmine, and granisetron. Please update the table to include these and any other relevant recent advancements.
2, The number of references provided is insufficient to fully explore recent progress in transdermal drug delivery. I suggest adding more references to enhance the content and provide readers with a more comprehensive understanding of the field.
3. Sections 5 and 6 primarily focus on microneedles. To enhance reader comprehension, it is recommended that an illustration of microneedles be included.
4. Please provide a summary of recent clinical trials related to transdermal drug delivery. Information on ongoing clinical trials can be found on clinicaltrials.gov, which will provide readers with the latest developments in this field.
Overall, these revisions will enhance the quality and comprehensiveness of this review and provide readers with a more informative and up-to-date analysis of recent advancements in transdermal drug delivery.
Author Response
- Thank you for your kind suggestions. We have included more transdermal products e.g. asenapine, donepezil, rivastigmine, granisetron, oxybutynin, norethindrone estradiol, buprenorphine, dextroamphetamine (approved in 2023), lidocaine as well as their application duration in Table 2.
-
We have added more references, giving more details on the pharmacokinetic profiles, recent findings from clinical trials of transdermal drugs and discuss more applications of transdermal patches (section 6.4 to 6.7). The total number of references now reach 156, compare to the previous 60 plus.
- Thank you for your kind suggestion. We have added a description on different types of microneedle in section 5, a new table (Table 3) and new figure (Figure 3).
-
The clinical studies have been added in Table 2 and discussed in section 6.4-6.7
Reviewer 2 Report
This review article is summarizing the recent advancement in transdermal patches. I appreciate researcher hard work to write this review article but author need word hard on it for publication. My comments are as follow;
General comments: This review article unable to provide the basic skeleton for review article. This manuscript far behind to match the journal standard. Researcher should summarize each section in a scientific way. For example, the section can be divided into. (1) Introduction, (2) Design of transdermal patches that contains basic component and type of transdermal patches such as dissolvable and biodegradable patches, (3) Applications that will include recent advancement in different diseases treatment, and (4) Conclusion and future direction.
Specific comments: Author should focus on the specific advancement in recent time either on design or in the applications.
1. Author can follow the (https://doi.org/10.1007/s13346-022-01138-1) review article to improve the research quality of article.
2. In application section author should mention why transdermal patches are better than conventional drug delivery methods.
3. What is the novelty of this review article since we have several review article on recent advancement on medical patches?
Conclusion: Overall, the manuscript need extensive major revision. The quality and content should be improved before publications.
Author Response
- Thank you for your kind suggestion. We have expanded the number of papers covered in this manuscript. More clinical findings as well as the pharmacokinetics studies are being included (section 6.4-6.7) to provide in-depth analysis of transdermal drug delivery to the readers. Nevertheless, this paper has been cited in the transdermal patches for gene therapy section.
-
Thank you for your kind suggestion. we have discussed the benefits of transdermal patches compare to other routes of administration in the first paragraph of Introduction.
-
Most of the published review articles either focus on the medical patch design or laboratory research. The novelty of this review paper lies in the in-depth discussion of clinical findings as well as the pharmacokinetics of these medical patches. We believe this will give readers more information and up-to-date analysis of recent advancements in transdermal drug delivery. In fact we also added the discussion on recently approved transdermal patches such as asenapine, donepezil, rivastigmine, and granisetron, which are not covered in other review articles.
Round 2
Reviewer 2 Report
The manuscript quality has been improved significantly and can be accepted in present form.